



# The spatiotemporal relationship between PM$_{2.5}$ and AOD in China: Influencing factors and Implications for satellite PM$_{2.5}$ estimations by MAIAC AOD

Qingqing He [1,2], Mengya Wang[3] and Steve Hung Lam Yim[3,4,1,*]

[1]Institute of Environment, Energy and Sustainability, The Chinese University of Hong Kong, Hong Kong, China

[2]School of Resource and Environmental Engineering, Wuhan University of Technology, Wuhan, 430070, China

[3]Department of Geography and Resource Management, The Chinese University of Hong Kong, Hong Kong, China

[4]Stanley Ho Big Data Decision Analytics Research Centre, The Chinese University of Hong Kong, Hong Kong, China

*Correspondence to*: Steve H. L. Yim (yimsteve@gmail.com)

**Abstract.** Satellite aerosol retrievals have been a popular alternative to monitoring surface PM$_{2.5}$ concentration due to its extensive spatial and temporal coverage. Satellite-derived PM$_{2.5}$ estimation strongly relies on an accurate representation of the relationship between ground PM$_{2.5}$ and satellite aerosol optical depth (AOD). Due to the limitation of satellite AOD data, most studies examined the relationship at a coarse-resolution (i.e., $\geq$ 10km) scale; more effort is still needed to better understand the relationship between 'in-situ' PM$_{2.5}$ and AOD at finer spatial scales. While PM$_{2.5}$ and AOD could have obvious temporal variations, few studies have examined the diurnal variation in their relationship. Considerable uncertainty therefore still exists in satellite-derived PM$_{2.5}$ estimation due to these research gaps. Taking advantage of the newly released fine-spatial-resolution satellite AOD data derived by the Multi-Angle Implementation of Atmospheric Correction (MAIAC) algorithm and real-time ground aerosol and PM$_{2.5}$ measurements, this study explicitly explored the relationship between PM$_{2.5}$ and AOD and its plausible impact factors including meteorological parameters and topography in mainland China during 2019, at various spatial and temporal scales. Coefficient of variation, Pearson correlation coefficient and slope of linear regression model were used. Spatially, stronger correlations mainly occurred in northern and eastern China and linear slope in northern inland regions was on average larger than those in other areas. Temporally, the PM$_{2.5}$-AOD correlation peaked in the noon and afternoon and reached the maximum in winter. Simultaneously considering relative humidity (RH) and planetary boundary layer height (PBLH) in the relationship can improve the correlation but the effect of RH and PBLH on the correlation varied spatially and temporally, both in strength and direction. In addition, the largest correlation occurred at 400-600 m primarily in basin terrain such as Sichuan Basin, Shanxi-Shaanxi Basins and Junggar Basin. MAIAC 1-km AOD can better represent the ground-level fine particulate matter in most domains with exceptions such as in very high terrain i.e. Tibetan Plateau and north-central China i.e. Qinghai and Gansu. Findings in this study have useful implications



for satellite-based PM$_{2.5}$ monitoring and will further inform the understanding of the aerosol variation and PM$_{2.5}$ pollution status in mainland China.

# 1 Introduction

Aerosols play an important role in regional and global climate change through the direct and indirect effects (King et al.,
1992;Li et al., 2007;Liu et al., 2019b;Liu et al., 2020;Liu et al., 2018). Fine particulate matter (PM$_{2.5}$) has attracted loads of concern from scientists, policymaker and the public due to its negative impact on the environment (Li et al., 2020;Yang et al., 2018b;Yim et al., 2019a) and human health (Schwartz, 1996;Pope III et al., 2002;Gu et al., 2018;Gu and Yim, 2016;Gu et al., 2020;Hou et al., 2019;Shi et al., 2020;Yim et al., 2015) . With an extended spatial and temporal coverage, the retrieval of surface PM$_{2.5}$ concentration from satellite aerosol optical depth (AOD) has become a popular approach to bridging the gap
left by ground-level monitoring network, facilitating the detection of the large-scale and long-term aerosol loading and their transboundary transport and the determination of the population exposure level for epidemiological and health study (Liu et al., 2017;Zou et al., 2019). The theoretical fundamental lies on the strong link between satellite AOD retrievals and ground-level PM$_{2.5}$ concentrations (Wang and Christopher, 2003). Therefore, it is of great importance to examine the PM$_{2.5}$-AOD relationship, not only for the relationship itself, but also for the underlying physical understanding involved.


With the rapid urbanization and industrialization, China has been suffering from serious air pollution (Zhao et al., 2019;Guo et al., 2011;He et al., 2019). Given the heavily haze pollution in China (Luo et al., 2018;Tong et al., 2018b, a;Yang et al., 2019b;Yim, 2020;Yim et al., 2019b), observation studies have demonstrated and investigated the PM$_{2.5}$-AOD relationship in China (Wang et al., 2014;Guo et al., 2017;Guo et al., 2009;Kong et al., 2016;Shao et al., 2017;Xin et al., 2016;Yang et al.,
2019a). For instance, the Moderate Resolution Imaging Spectroradiometer (MODIS) Dark Target (DT) and Deep Blue (DB) 10-km AOD products have been widely used for quantifying the PM$_{2.5}$-AOD relationship in both regional (Kong et al., 2016;Shao et al., 2017) and national scales (Guo et al., 2017;Yang et al., 2019a). The relationship is found not always tuning and exhibits large spatiotemporal variability. With the development of satellite remote sensing technique, a global daily aerosol product at a high spatial resolution, i.e., the Multi-Angle Implementation of Atmospheric Correction (MAIAC) 1-km
AOD product has been released recently, providing local-scale aerosol gradients and details missed by previous coarse-resolution (10 km) products (Lyapustin et al., 2018). Because the ground-level PM$_{2.5}$ concentration is generally measured at an in-situ point while the satellite AOD is reported at a grid (e.g., with 1 km × 1 km for MAIAC AOD and 10 km × 10 km for DT AOD), the relationship between a "point" PM$_{2.5}$ and a "gridded" AOD at a coarse resolution might mask their discrepancy or correspondence at a finer resolution. That is, due to the presence of characterizing fine-resolution aerosol
heterogeneity as well as the changes in retrieval scheme and accuracy (Guo et al., 2017), the use of the high-resolution AOD data probably introduces different/additional uncertainties into the PM$_{2.5}$-AOD association, leading to a different



spatiotemporal pattern and further affecting the performance of satellite-derived $PM_{2.5}$ prediction. Thus, further exploring the relationship between near-surface $PM_{2.5}$ and MAIAC AOD for China during a long-term period is necessary.

The $PM_{2.5}$-AOD link has been widely suggested to vary in space and time, which is associated with the different definitions between surface $PM_{2.5}$ measurement and satellite AOD involved in aerosol types, meteorological condition, and topography (Kloog et al., 2014;Lee et al., 2016;Ma et al., 2014;Wang et al., 2014;Zhang and Li, 2015) . The ground-level air quality monitoring equipment contains a dynamic heating system to evaporate water vapor in particles and thus measures dry mass concentration for $PM_{2.5}$ concentration, which is inevitably different with satellite AOD that interprets aerosol extinction

without considering the humidity condition. The vertical structure can induce uncertainty between $PM_{2.5}$ and AOD because satellite AOD is an integration of aerosol extinction in the entire atmosphere while ambient $PM_{2.5}$ concentration is only measured at the ground surface. Irrespective of different conditions, the meteorological factors may have confounding influence, causing the relationship to a different extent and even leading to a contrary result (i.e., improvement or deterioration). In addition, topography is another major contributor to the uncertainty existed in the $PM_{2.5}$-AOD relationship

but the impact is difficult to be quantified due to its complex structure. Therefore, it deserves further explicit analyses to explore the effect of terrain and meteorological factors, including hygroscopicity and vertical structure of aerosols on $PM_{2.5}$-AOD relationship, especially for the mainland China with a complex terrain and land cover types.

The diurnal cycle of surface $PM_{2.5}$ concentration has various applications, e.g., the short-term health impact and aerosol-

cloud interaction (Arola et al., 2013;Guo et al., 2017). To successfully capture the diurnal variation in the $PM_{2.5}$-AOD relationship and represent the diurnal cycle of $PM_{2.5}$, understanding the diurnal variation involved in the data is crucial. However, it is easily ignored in studying the relationship between surface $PM_{2.5}$ and satellite AOD due to the limitation of hourly satellite aerosol observations and the difficulty in making hourly $PM_{2.5}$-AOD collocations. To the best of our knowledge, few studies have taken into account the diurnal cycle in the data when investigating the $PM_{2.5}$-AOD relationship.


Therefore, this study aimed at comprehensively investigating the spatiotemporal relationship between ground-level $PM_{2.5}$ and satellite AOD in the mainland China using the MAIAC 1-km aerosol product with multi-scale analyses in space and time, with an emphasis on the difference in the spatial variation and temporal trend against previous relationships dependent on coarse-resolution data. We carried out correlation analyses measured by Pearson correlation coefficient and slope of

linear regression to explore the $PM_{2.5}$-AOD relationship at 1500 monitoring stations, in more than 360 cities and eleven urban agglomerations, by taking the latent impact of diurnal variation of aerosol particle, relative humidity (RH), planetary boundary layer height (PBLH) and elevation into account. In addition to monthly variation, seasonality and annual pattern, the real-time ground aerosol observations were also employed in this study to uncover the diurnal variability and impact on $PM_{2.5}$-AOD relationship using hourly comparisons during a day when Terra and Aqua data are available. The remaining of

this study is organized as follows: descriptions of the data used, including the ground-level $PM_{2.5}$ measurements, the MAIAC



AOD, RH, PBLH and elevation in China, are given in Section 2. The results regarding correlation and linear slope analysis in PM$_{2.5}$-AOD association, and its impact factors are presented in Section 3 and discussed in Section 4. The major conclusions are summarized in the final section.

## 2 Materials and Methods

### 2.1 Study area and urban agglomerations

The study area is the mainland China (excluding Taiwan, Hong Kong and Macau), covering more than 360 cities and eleven urban agglomerations. The eleven urban agglomerations include five national urban agglomerations, i.e., the Beijing-Tianjin-Hebei (BTH), Yangtze River Delta (YRD), Pearl River Delta (PRD), Triangle of Central China (TCC), and Chengdu-Chongqing (CDCQ) urban agglomerations, and six regional urban agglomerations, i.e., the central-and-southern Liaoning (CSL), Shandong Peninsula (SDP), Harbin-Changchun (HC), Central Plain (CP), Guangzhong Plain (GZP), Northern Tianshan (NTS) urban agglomerations. According to the "National New-type Urbanization Plan (2014-2020)" (http://www.gov.cn/zhengce/2014-03/16/content_2640075.htm), these agglomerations are densely populated and have relatively well-developed urban systems. The locations and cities for these agglomerations are detailed in Fig. A1 in Appendix A.

### 2.2 Data collection and preprocessing

Hourly ground-level PM$_{2.5}$ measurements from ~1500 monitoring stations for 2019 were collected from the official website of the Ministry of Ecology and Environment of China (https://www.mee.gov.cn). These observed PM$_{2.5}$ mass concentrations are measured by the tapered-element oscillating microbalance or beta-attenuation method with quality control (National Ambient Air Quality Standards, GB3095-2012; http://english.mee.gov.cn/Resources/standards/Air_Environment/quality_standard1/201605/t20160511_337502.shtml).
Hourly concentrations with <1 μg/m$^3$ were discarded because they are below the monitors' limit of measurement. Daily PM$_{2.5}$ values were averaged from hourly concentrations during 0:00-23:00 local time and those with at least 75% valid hourly observations were remaining for statistical analyses.

The MODIS 1-km AOD data archived in Collection 6 derived by the MAIAC algorithm in 2019 over China were used in this study. This new aerosol dataset not only represents aerosol retrievals at high spatial resolution (Chudnovsky et al., 2013;Lyapustin et al., 2018) but also achieves a comparable accuracy with the previous popular coarse-resolution aerosol product (Zhang et al., 2019;Liu et al., 2019a;Martins et al., 2017), making it possible to detect local aerosol variability. Both Terra (passing over at 10:30 am local time) and Aqua (crossover at 1:30 pm local time) MAIAC AOD data were downloaded from the NASA Level-1 and Atmosphere Archive and Distribution System (LAADS) Distributed Active Archive Center





(DAAC). To investigate the aerosol diurnal variation, the ground-level AOD observations from Version 3.0 Aerosol Robotic Network (AERONET) with assured quality were obtained.

Meteorological variables including RH and PBLH and terrain were used for impact factor analysis. The RH data were
obtained from the China Meteorological Data Service Center (https://data.com.cn) and the PBLH data were downloaded from the reanalysis dataset provided by the European Center for Medium-Range Weather Forecast (ECMWF, https://www.ecmwf.int/). The Advanced Spaceborne Thermal Emission and Reflection Radiometer (ASTER) Global Digital Elevation Model (GDEM) was collected to measure the terrain in terms of elevation.

## 2.3. Data collocation and analytical methods

According to previous studies Liu et al. (2019a), the MAIAC AOD data also confront the missing data problem as its coarse spatial resolution. To tackle this issue and improve the number of daily collocations with ground $PM_{2.5}$ measurements, an imputation method was conducted to fill the missing AOD values for one satellite when the other was available, and the improved AOD data served as the daily mean AOD for the collocation with daily $PM_{2.5}$ averages. The imputation is similar to our previous studies (He et al., 2020, 2019), through constructing a relationship between Terra and Aqua retrievals for
each observation day and applying the obtained coefficients to predict the missing values. In addition, hourly collocation was also accomplished to support the hourly analyses in this study, which directly collocated the hourly $PM_{2.5}$ measurements at each observational site with the Aqua/Terra AOD retrievals within the hour of its overpass. All the daily and hourly collocations required the centroid of AOD was spatially matched the coordinate of the ground-level monitors.

The coefficient of variation (CV), defined as the ratio of the standard deviation to the mean to measure the dispersion of data points around the mean, allows us to compare the degree of difference between ground $PM_{2.5}$ and columnar AOD. The relationship between ground $PM_{2.5}$ and satellite AOD was quantified by Pearson correlation coefficients (r). The linear regression model was applied using AOD as an independent variable and $PM_{2.5}$ a dependent variable. The estimated slopes of the linear models were used for the statistical analysis.


To examine the impact of hygroscopicity and vertical structure of aerosols on $PM_{2.5}$-AOD relationship, the RH correction and vertical correction were applied to obtain RH-corrected $PM_{2.5}$ and PBLH-corrected AOD for the subsequent correlation analyses. Following previous studies (Kong et al., 2016; Wang et al., 2014), the RH-corrected $PM_{2.5}$ and PBLH-corrected AOD were calculated as follows, respectively:

$$Corrected\ PM_{2.5} = PM_{2.5}(1 - \frac{RH}{100}) \tag{1}$$

$$Corrected\ AOD = \frac{AOD}{PBLH} \tag{2}$$



## 3 Results

### 3.1 Characteristics of variation in surface PM$_{2.5}$ and satellite AOD

Figure 1 illustrates the annual mean PM$_{2.5}$ from in-situ data vs. AOD from the entire satellite observed grid cells throughout the mainland China in 2019. The typical hotspot polluted by fine particles could be identified in the North China Plain (NCP), in particular in southeastern Hebei, western Shandong and northern Henan provinces with PM$_{2.5}$ concentration up to 55 μg/m$^3$ or larger. It is noted that southern and western Xinjiang, the margin of Taklimakan Desert, had very high annual concentration exceeding 75 μg/m$^3$, highlighted as one PM$_{2.5}$ pollution center. In general, the MAIAC AOD shared a similar spatial pattern with PM$_{2.5}$. That was, areas with high AOD values congruously correspond to those with high PM$_{2.5}$ concentration except for typical mismatch in southwestern China. For example, higher mean AOD with values of 0.5-0.8 is found in Chengdu-Chongqing (CDCQ) agglomeration, which was as high as that in the NCP, contrary to the relatively low PM$_{2.5}$ concentration with a mean of 38.93 μg/m$^3$ (mean concentration for NCP was 49.32 μg/m$^3$).

[Figure 1 is about here]

Figure S3 shows the temporal profile of PM$_{2.5}$ and AOD in the entire country in 2019. The national PM$_{2.5}$ average was 38.53 μg/m$^3$, and the annual mean AOD was 0.22. The monthly changes of the two indicators show a different temporal pattern. The monthly PM$_{2.5}$ concentration shows a U-shape trend, which remained low and stable between June and August, and the monthly concentration in summer months was almost a third of those in winter months. This result is consistent with findings reported in previous researches (Yang et al., 2018a). In contrast, the monthly AOD variation shows a rough inverse U-shape curve with the plateau between March and August. The amplitude of monthly AOD variation was also less than that of PM$_{2.5}$: the maximum AOD level of 0.28 was occurred in March, which was 0.1 (~35.42%) higher than to the smallest AOD month in November. This further demonstrates the difference between ground-level PM$_{2.5}$ and satellite AOD observations in Figure 1.

The diurnal characteristics of PM$_{2.5}$ variations in China in the 2019 were investigated. In general, the national maximum PM$_{2.5}$ diurnal mean concentration occurred in morning and evening, whereas the minimum mean values appeared in afternoon (Figure S4), generally consistent with the diurnal pattern revealed in previous studies (Guo et al., 2017). The two peaks appeared in the overall diurnal PM$_{2.5}$ curves were linked to PM$_{2.5}$ spatial variation. Figure 2 shows the spatial distribution of the hours at which the maximum and minimum PM$_{2.5}$ concentration occurred and the corresponding concentration value. Among the 1494 monitoring stations, the largest diurnal PM$_{2.5}$ concentration appeared in the evening (19:00-22:00) at 559 sites (~37.42%), followed by 519 sites (~34.74%) with the maximum concentration in morning (7:00-10:00), and there were only 50 sites (~3.35%) reaching its maximum in afternoon (12:00-18:00). To be more specific, the



significant spatial variability in the highest diurnal $PM_{2.5}$ concentration can be found. The morning $PM_{2.5}$ peak generally
      occurred in the NCP while the evening maximum prevailed in the southern and northeastern China where the afternoon
      $PM_{2.5}$ maximum sporadically appeared. Besides, the highest diurnal mean concentration in morning peak was generally
      greater than that in the evening, at an average of 46.60 μg/m$^3$ vs. 42.24 μg/m$^3$. Among the 413 sites in NCP, there were more
      than 94% sites having concentration exceeding 35 μg/m$^3$, pinpointing the serious $PM_{2.5}$-pollution risk in the NCP region. We

note that there was almost not any spatial difference in the minimum hourly mean concentration, which uniformly appeared
      between 14:00 and 17:00.

      [Figure 2 is about here]

The CV analysis was conducted based on temporally averaged $PM_{2.5}$ and AOD values to evaluate the spatial variation of
      aerosol particles over a given region. As shown in the upper panel of Figure 3, the ambient $PM_{2.5}$ concentration in 2019 over
      the CP region was the highest (multi-year mean value of 57.69 μg/m$^3$) associated with a moderate spatial variability (CV of
      0.79), suggesting that the CP region was the most polluted among the 11 urban agglomerations and the majority of cities
      within this area were exposed to very dangerous $PM_{2.5}$ levels. With the lowest $PM_{2.5}$ concentration (27.43 μg/m$^3$) as well as

the smallest CV value (0.58), almost all PRD cities experienced relatively clean ambient air. The $PM_{2.5}$ concentration in NTS
      urban agglomeration remained a moderate level (45.96 μg/m$^3$), but the CV value was the largest (CV=1.25), implying a
      distinct contrast in air quality between cities over the region. Figure 3 also presents a different regional pattern in both mean
      AOD and CV values.

[Figure 3 is about here]

**3.2 Spatiotemporal variability of $PM_{2.5}$-AOD relationship**

      Figure 4 presents the spatial distribution of correlation coefficients (upper panel) and linear trend (lower panel) between the
      daily mean $PM_{2.5}$ concentration and AOD in the entire study period and the four seasons over all monitoring sites. With
      correlation coefficient (r) values ranging from -0.5 to -0.18, abnormal negative associations between $PM_{2.5}$ and AOD were

observed in the Tibetan Plateau (TP) where the altitudes are almost higher than 4000m. In addition to TP, the correlation
      coefficient for each monitoring site ranged from 0.15 to 0.84. In general, the stronger correlations (r>0.4) were mostly
      concentrated in the northern and eastern China, whereas r values for most observational sites dropped below 0.4 in southern
      areas.

Particularly, Sichuan Basin (SCB) was highlighted in Figure 4 (a) with the strongest correlation (r >0.6), whereas very low
      correlations (r<0.3) were found in northern Xinjiang and north-central China. Fig 4 (b-e) exhibits a distinct seasonal
      variability of $PM_{2.5}$-AOD even though the spatial pattern in each season is generally similar to that shown by all data pairs in



Figure 4 (a). The overall correlation coefficient in winter was relatively higher, with overall r of 0.61, which demonstrates that the light extinction in the atmosphere in wintertime may be mainly caused by fine particles near the surface, then

followed by spring (0.57) and autumn (0.55). The r value reached the lowest in summer (0.47). The distribution of correlations along with various $PM_{2.5}$ levels may account for the seasonal difference. As shown in Figure S5, correlations between $PM_{2.5}$ and AOD were the strongest in extremely high $PM_{2.5}$ concentration levels, and coincidentally, higher $PM_{2.5}$ concentrations typically occurred in winter, thus leading to a better $PM_{2.5}$-AOD relationship in this season. Also, the seasonal variation was probably related to lower RH in winter and unstable atmosphere condition in summer.


Figure 4(f-j) shows the spatial distribution of slopes (S) of linear regressions between daily $PM_{2.5}$ and AOD for each monitoring site. Corresponding to the correlation analysis in Figure 4 (a-e), the slope values in TP were also negative, and for the other area over the mainland China, the slopes of linear regressions ranged from 9.37 to 158.45 over the entire observation period (Figure 4 (f)) with significant spatial and seasonal variabilities (Figure 4 (g-j)).


[Figure 5 is about here]

To better represent the spatiotemporal features, correlation analyses between ground-level $PM_{2.5}$ and satellite AOD have been conducted over all monitoring stations exhibited in Figure 4, through zooming in to urban-agglomeration scale in

Figure 5. In general, the $PM_{2.5}$-AOD relationships at the regional scale mirrored those at the observational-site scale. In terms of correlation coefficient, it reached higher values in urban agglomerations located in northern and eastern China such as the GZP, BTH and CDCQ agglomerations with r values greater than 0.50; the weakest relationship with r of 0.26 occurred in the NTS agglomeration. As for the slope of the regression function, the maximum slope (81.10) was observed at the GZP region, followed by the CSL (51.48) and BTH (51.37) agglomerations; the lowest value was at the PRD urban agglomeration with a

slope of 23.00.

Figure S7 demonstrates that the spatial distribution of $PM_{2.5}$-AOD correlations during the study period using Terra and Aqua AOD matched with $PM_{2.5}$ measurements at respective overpass hour. It is intriguing that correlations for almost all monitoring sites at 2 pm were greater than those at 11 am, and the overall correlation at 2 pm was 0.54, which was also

obviously higher than that of 0.47 at 11 am. To determine whether the difference is caused by aerosol retrieval accuracy from the Terra (i.e. at 11 am) and Aqua (i.e. at 2 pm), we conducted a comparison analysis using hourly ground $PM_{2.5}$ and the spatiotemporal collocated AERONET AOD observations. Figure S8 indicates that the diurnal correlations between ambient $PM_{2.5}$ and AERONET AOD were also larger in afternoon.



### 3.3 Influence of meteorology on PM$_{2.5}$-AOD correlation

To examine the spatiotemporal effect of RH and PBLH on daily PM$_{2.5}$-AOD relationship, we conducted correlation analyses using RH-corrected PM$_{2.5}$ data and PBLH-corrected AOD data for monitoring sites (Figure 6) and for the 11 agglomerations (Table 1). The details of the correction method for PM$_{2.5}$ and AOD have been provided in Section 2.3. In general, after correction by RH and PBLH (Figure 6), the mean correlation coefficient r reached 0.71, and the r values in ~ 16% sites increased to >0.80, which presents a notable improvement in PM$_{2.5}$-AOD relationship compared to those based on

uncorrected data (mean r=0.44 and the proportion of >0.80 is 0.25%). For individual observational sites, ~96% of monitoring sites had an enhanced PM$_{2.5}$-AOD correlations after both corrections, increasing by 0.01~0.61. The correlation after RH correction slightly increased by an average of ~0.07, and the vertical correction achieved a better improvement on PM$_{2.5}$-AOD relationship than the former, which contributed to the average r increase by ~0.18. Specifically, ~22% of monitoring sites, primarily located in Guangdong, Guangxi, Yunnan, and Heilongjiang provinces, had larger r values after RH

correction than vertical correction. These results indicate that the spatial difference existing in the influence of RH and vertical correction on the relationship between PM$_{2.5}$ and AOD cannot be ignored, due to their significant impact on the relationship in most monitoring sites. It should be noted that there was an obvious negative increment after both corrections, in particular in most monitoring stations in Yunnan province.

[Figure 6 is about here]

There was also significant seasonal discrepancy existed in the meteorological effect on improving PM$_{2.5}$-AOD relationship. As shown in seasonal patterns of Figure 6, after both corrections, the correlations had the largest improvement in autumn with the mean r value increased by 0.21, followed by spring (0.11) and winter (0.10), and summer had the smallest

improvement with an increase by 0.09. Specifically, in spring and autumn, both RH and vertical corrections imposed a positive effect on PM$_{2.5}$-AOD correlation, but the strength was different. In spring, RH correction had a greater contribution to improving PM$_{2.5}$-AOD correlation compared with vertical correction, and the r value based on RH-corrected data was 0.02 larger at an average level. In autumn, vertical correction exerted a more important influence than RH correction, with the mean r value of 0.68 vs. 0.65. It should be noted that the correlation revised by PBLH was lower than the original condition

in summer with the mean r value reducing by 0.05, especially for monitoring stations in central China such as Jiangxi, Shaanxi, and Henan provinces where significant decreases in r values (changing of <-0.15) occurred.

[Table 1 is about here]

Table 1 illustrates the statistics of PM$_{2.5}$-AOD relationship based on both corrected and uncorrected data over the 11 urban agglomerations. It follows that there are also obvious spatial and seasonal discrepancies in the influence of humidity and



vertical distribution on regional PM$_{2.5}$-AOD association. In summer, the correlation coefficient r of AOD, vertically revised by PBLH, and ground PM$_{2.5}$ had a significant decrease over the original relationship, except for the NTS and YRD regions. Despite of a general weak relationship in the NTS region, PBLH-corrected AOD was superiorly related to surface PM$_{2.5}$

observations than the original correlation, not only in summer (0.29 vs. 0.21), but also in spring (0.42 vs. 0.37) and autumn (0.45 vs. 0.23).

### 3.4 Impact of terrain on PM$_{2.5}$-AOD correlation

The effect of topography on the PM$_{2.5}$-AOD correlation analyses was investigated by separating the collocated samples into 21 elevation bins with an interval of 200 m. As shown in Figure 7, a distinct trend and pattern can be identified between terrain and PM$_{2.5}$-AOD relationship. Corresponding to the abnormal negative relationship over Tibetan areas found in Figure

4, the correlation coefficients between PM$_{2.5}$ and AOD at a higher elevation, namely, greater than ~ 3200 m, were also less than zero. The correlation peaked at 400-600 m with r of 0.60 and p-level<0.05. These samples were then overlaid onto the spatial map of topography (Figure S9) and found that those having the maximum r values were primarily located in basin terrain such as Sichuan Basin, Shanxi-Shaanxi Basins and Junggar Basin.


[Figure 7 is about here]

### 4 Discussion

Our results show a mismatch between AOD values and PM$_{2.5}$ in southwestern China. This was probably attributed to the different available observations between the two metrics of aerosol loading: AOD derived from satellite only offered

relatively fewer effective retrievals over this area due to cloud contamination (Figure S2) while surface PM$_{2.5}$ observation has almost no such limitation. Consequently, the limited satellite AOD samples cannot fully represent the actual aerosol loading. If those missing AODs are not filled in, excessive uncertainty will be induced to the subsequent PM$_{2.5}$-related health research. Thus, a particular caution should be paid to the mismatch between satellite AOD and ground PM$_{2.5}$ caused by the amount of observations.


We found that mean AOD and CV values had a different regional pattern. This result suggests that SDP was the uniformly polluted by aerosol particles throughout the atmosphere because the largest mean AOD value (0.48) was observed in this region with the smallest CV value (0.16). In addition, the regional CV in relation to PM$_{2.5}$ generally exhibited larger values than AOD, revealing that the extent of spatial non-stationarity shown by PM$_{2.5}$ was obviously stronger compared to AOD in

each individual region. Notably, the national CV of AOD was remarkably larger than the regional ones and contrarily higher than the PM$_{2.5}$ CV, implying that the overall degree of spatial variability represented by ground-level PM$_{2.5}$ measurements was weaker than that of satellite AOD at a national scale. It is probably due to the fact that the monitoring stations only sparsely located in urban areas while satellite AOD can almost cover the whole study domain. The PM$_{2.5}$ measurements





represent the pollution level at a particular location while the satellite AOD interprets the average extinction level for 1-km
"area". These further corroborate the mismatch between PM$_{2.5}$ and AOD pattern observed in Figure 1 and Figure S3.

For the spatiotemporal variability of PM$_{2.5}$-AOD relationship, Tibetan Plateau shows abnormal negative associations
between retrieved PM$_{2.5}$ and AOD. According to Lyapustin et al. (2018), the MAIAC algorithm did not make reliable aerosol
retrievals at very high elevations, and a fixed climatology value should be applied instead, thereby leading to the PM$_{2.5}$-AOD
relationship over TP contrary to other areas. While we found a large spatial discrepancy of correlation coefficient for each
monitoring site that was consistent with previous studies using MODIS DT aerosol data (Guo et al., 2017;Yang et al.,
2019a), the relative low r values (<0.4) in southern areas were slightly different from previous studies (Guo et al., 2017).
Based on MODIS DT AOD dataset, Guo et al. (Guo et al., 2017) reported that correlation coefficients between PM$_{2.5}$ and
AOD in eastern China were apparently larger than those in the western China because DT algorithm had a larger uncertainty
in the western arid and semi-arid areas. Unlike DT algorithm that can only derive reliable AOD values over surface with low
albedos, MAIAC could provide aerosol retrievals in both bright and dark targets. As shown in Figure S2, MAIAC aerosol
product can provide more retrievals over the northern arid and semi-arid areas than southern China. Moreover, according to
(Liu et al., 2019a), the accuracy of MAIAC AOD in unoccupied land, built-up and mixed regions was evidently better than
that by DT. These above-mentioned evidences in turn result in relatively lower biases for the PM$_{2.5}$-AOD relationship under
these conditions and thus partly account for the difference of spatial pattern in PM$_{2.5}$-AOD association between using
MAIAC and DT AOD products. In addition to the accuracy and data availability of aerosol retrieval algorithms, the natural
condition including climate and terrain also had obvious influences on the relationship between PM$_{2.5}$ and AOD. For
example, compared with southern China, the RH in the north was generally lower, which contributed to the harmony in the
PM$_{2.5}$-AOD association in terms of humidity condition; in addition, monsoon could cause aerosols unstable in subtropical
climate zone, which may weaken correlations in these areas.

Obvious a spatial variation of PM$_{2.5}$-AOD correlation and slopes of linear regressions between daily PM$_{2.5}$ and AOD for each
monitoring site were observed in our results. The largest correlation in Sichuan Basin was most likely due to its bowl
topography. The surrounding terrain is not favorable for pollutant dispersion and thus causes pollutant acumination in the
local environment (Ning et al., 2018;Ning et al., 2019;Ning et al., 2020). This explains a good agreement between ground-
level PM$_{2.5}$ and total column AOD. For the low correlation in north-central China, e.g. Qinghai and Gansu provinces,
MAIAC AOD retrievals were replaced with a mean value over a mesoscale area of 150 km (Lyapustin et al., 2018), which
cannot resolve the 1-km aerosol variability as elsewhere and thus bring additional biases into the correlation between PM$_{2.5}$
and AOD, thereby leading to incongruous associations between PM$_{2.5}$ and AOD over these areas. Regarding the slopes of
linear regression between daily PM$_{2.5}$ and AOD, a larger slope suggests less aerosol extinction. The higher S values in inland
areas and during winter indicate that, under these circumstances, AOD was not very sensitive to PM$_{2.5}$ change. This was
probably attributed to the spatial discrepancy and seasonal variation in a humid condition. Due to the increased



hygroscopicity, the size of aerosol particles became larger than dry particles (Silva et al., 2015), which allowed more light to be scattered in the context of higher RH, e.g., over the coastal areas and during summer. In addition to humid condition, aerosol type was another reason to account for the spatial variability in slope of PM$_{2.5}$-AOD relationship. Previous studies have borne out that the regression function between ambient PM$_{2.5}$ concentration and AOD corresponded with the typical air masses over a specific region (Kong et al., 2016), and aerosol extinction property cannot be as sensitive in areas dominated by coarse-mode aerosols (Kong et al., 2016;Xin et al., 2014). As indicated by the spatial distribution of mean Angström exponent (AE or α) in Figure S6, over China, aerosol particles with a larger size (smaller AE) such as dust primarily dominated in northern areas while the fine-mode aerosols prevail in the south, thereby leading to a greater slope of the PM$_{2.5}$-AOD relationship in northern China. It should be noted that the regional pattern of PM$_{2.5}$-AOD relationship was a little different from the pattern by individual monitors because the regional association was an integration of observational sites within an urban agglomeration other than a simple averaging of all sites within the region. Moreover, we found that most of regional correlations based on MAIAC AOD in this study were closed to (and even slightly higher than) those by MODIS DT AOD in previous studies (Guo et al., 2017;Yang et al., 2019a). For example, the correlation over the BTH, PRD, SDP, and CP region was 0.55, 0.38, 0.42, and 0.39, respectively, while the corresponding correlation in this study was 0.53, 0.30, 0.48, and 0.47, respectively. This was because MAIAC AOD (1 km spatial resolution) had a refined spatial resolution than DT AOD (10 km), thereby reducing the variability in aerosol inside a grid cell collocated with the in-situ measurements. The improved overall retrieval accuracy performed by MAIAC algorithm (Zhang et al., 2019;Liu et al., 2019a) may be another reason, which can reduce the biases for the correlation analysis in PM$_{2.5}$-AOD relationship. These findings suggest that MAIAC AOD product is a better indicator of fine particle pollution compared with MODIS coarse-resolution aerosol product.

We also found significant effect of meteorology on PM$_{2.5}$-AOD correlation using Terra and Aqua AOD. Previous study has evaluated the MAIAC retrieval performance of Terra and Aqua AOD datasets against ground-level aerosol observations and found that AODs from the two satellites presented a similar retrieval accuracy (Zhang et al., 2019). Thus, we deduced that the difference in correlation between the two times (11 am & 2 pm) was most likely due to the diurnal variability of meteorology, e.g., PBLH. The diurnal cycle of PBLH was more likely one important factor leading to the noon and afternoon maximum. It is because higher PBLH often observed in the noon and afternoon would definitely allow air pollutants to be mixed vertically such that the PM$_{2.5}$-AOD relationship should be closer. At nighttime, the relative stable atmosphere may be favorable for the accumulation of aerosol particles near the ground surface, and thus AOD may not be able to reflect the ground-level aerosol concentration as well as they do in daytime. Our results show a significant improvement on PM$_{2.5}$-AOD correlation after RH and vertical correlation, suggesting the importance of applying the correction for the retrieval work. It should be noted that there was an obvious negative increment after both corrections, in particular in most monitoring stations in Yunnan province. The poorer correlation may be attributed to the vertical correction, because the single correction by humidity can slightly enhance PM$_{2.5}$-AOD correlation that contributed to the r



values rise by an average of 0.01 but the correlations revised by vertical data remarkably reduced by 0.04~0.42 for individual sites. Due to the East Asian and Indian monsoons, extraneous aerosols such as smoke and dust transported from India, Burma and desert regions in northern China can form elevated aerosol layers in the Yunnan-Guizhou Plateau (Zhu et al., 2017;Liu et al., 2016), thus leading to the local AOD vertically corrected by PBLH poorly related to the ground-observed $PM_{2.5}$ in this region. The effect of meteorology on $PM_{2.5}$-AOD relationship improvement was also found to vary seasonally. Nevertheless, our results show vertical correction in summer may not be useful to improve $PM_{2.5}$-AOD in central China such as Jiangxi, Shaanxi, and Henan provinces. Central China is located in a typical subtropical monsoon climate zone, the summer monsoon causes frequent occurrence of unstable weather and increases the dissipation of local aerosols associated with the scour effect of rainfall and atmospheric convection (Gong et al., 2017;Wang et al., 2011), which may account for the weaker $PM_{2.5}$-AOD correlation by PBLH correction. RH correction had a weaker influence in winter during which the averaged coefficient increased less than 0.01, which may be related to the dry climate in winter. These findings signify that seasonal variability in the data plays an important role on improving $PM_{2.5}$-AOD relationship, and cannot be ignored, despite of the existing differences in corrected correlations. On the other hand, PBLH-correction can effectively improve $PM_{2.5}$-AOD relationship in northwestern China. Contrary to the poor effect of vertical revision in summer and spring, seasonal $PM_{2.5}$ revised by humidity was more closely linked to satellite AOD than the uncorrected relationship in almost all agglomeration areas, except for summer and autumn in the CDCQ area and winter in the BTH and CP regions. This further corroborates that regional pattern was a little different from the pattern presented by monitoring stations, and the RH and vertical corrections did not improve the correlation between ground measured $PM_{2.5}$ and satellite AOD with significant scope but should be tailored for specific areas.

Besides meteorology, terrain was also found to have impact on $PM_{2.5}$-AOD correlation, especially in basin areas. It may be because basin terrain tends to form a local atmospheric environment, aerosol particles can mix well in the entire atmosphere. Moreover, the complex topography protects the basins from transboundary transport of air pollution (Ning et al., 2019;Ning et al., 2020). These two reasons contributed to the consistency between near-surface $PM_{2.5}$ and column AOD. Figure 7 also shows that the r values tended to become lower as the elevation rises, which indicates a deteriorated $PM_{2.5}$-AOD correlation with the elevated altitude, especially in the eastern China where the majority of observational sites are in operation. Due to the spatially uneven distribution of air quality monitoring stations, there were only 23.68% pairs of the total matched samples located at elevations higher than 1000 m, leading to Figure 7 and Figure S9 mainly representing the eastern-China effect of terrain on $PM_{2.5}$-AOD relationship.

At last, it is necessary to discuss the implication for ground $PM_{2.5}$ derived from satellite AOD. With the release of MODIS MAIAC aerosol product, the high-spatial-resolution AOD dataset has become an emerging application to inferring fine particulate matters in China. This study profiled the relationship between the ground-level $PM_{2.5}$ and the 1-km satellite AOD in the perspective of spatiotemporal variations and analyzed the underlying factors that contribute to the current pattern.



Corresponding to previous studies(Yang et al., 2019a;Shao et al., 2017;Wang et al., 2014), the PM$_{2.5}$-AOD association in this study also presents a significant spatiotemporal variation. The meteorological parameters and topography have a complicated influence on the PM$_{2.5}$-AOD relationship, posing extreme challenges for ground PM$_{2.5}$ estimations. For example, PBLH has an adverse impact on the relationship in Yunnan, whereas humidity may not be useful for improving the association in winter. Aerosol type is another contributor accounting for the uncertainties involved. However, as our results show, the pattern based on the 1-km MAIAC AOD is a little different from previous studies using coarse-resolution aerosol products such as the MODIS 10-km DT or DB AOD datasets (Guo et al., 2017;Yang et al., 2019a), which should be noticed for PM$_{2.5}$ predictions. For example, the aerosol retrieval scheme implemented by the MAIAC algorithm at very high altitudes brings about the unnatural negative PM$_{2.5}$-AOD correlation over Tibetan region so that it has a direct effect on PM$_{2.5}$ estimation ability. To clarify this issue, we established two separate multiple linear regression models using the same daily samples: one was for areas with elevation higher than 3200 m and the other was for the remaining area and obtained significant different R2 of 0.14 vs 0.41 at p-level < 0.01. This result suggests that the PM$_{2.5}$ prediction model should treat very high-altitude areas with caution; otherwise additional uncertainty within the data may hinder the acquisition of model parameters and further attenuate the prediction accuracy. Therefore, it should be keeping a discreet attitude to apply the newly released aerosol product to PM$_{2.5}$, especially for large areas with a complicated terrain and land cover.

## 5 Conclusions

In this study, surface PM$_{2.5}$ measurements in 2019 across China were spatially and temporally matched with MAIAC 1-km AOD, combined with humidity and vertical data and topography to perform explicit correlation analyses, which are the cornerstone for retrieving ground-level PM$_{2.5}$ from satellite aerosol loading. We found that, under the MAIAC high-resolution perspective, the relationship between surface PM$_{2.5}$ and column AOD was generally in a good agreement with previous results but also shows a little difference in spatiotemporal variability, which should be cautious for applications to both aerosol-related and PM$_{2.5}$-related studies. The findings in this study are helpful to better understand the PM$_{2.5}$ pollution status in China and also its relationship with column AOD, facilitating the reconstruction of high-resolution PM$_{2.5}$ from satellite aerosol retrievals.

The relationship between PM$_{2.5}$ and AOD varied notably in mainland China, both spatially and temporally, in terms of correlation coefficients and slopes of linear regression. Spatially, stronger correlations mainly occurred in northern and eastern China and linear slope in northern inland regions was on average larger than those in other areas. Temporally, the PM$_{2.5}$-AOD correlation peaked in the noon and afternoon and varied greatly in seasons, among which the maximum correlation happened in winter (r=0.61). MAIAC 1-km AOD can better represent the ground-level fine particulate matter in most domains with exceptions such as in very high terrain i.e. Tibetan Plateau and north-central China i.e. Qinghai and



Gansu, where MAIAC does not interpret the actual 1-km aerosol loading, resulting in unnatural or weaker $PM_{2.5}$-AOD relationships for these domains. This merits more attention in the application of $PM_{2.5}$ estimations.

As far as the meteorology on the relationship between $PM_{2.5}$ and AOD was concerned, we found that simultaneously considering RH and PBLH in the relationship could improve the correlation but the effect of RH and PBLH on the relationship varied spatially and temporally, both in strength and direction. The diurnal cycle of PBLH coincided with that of $PM_{2.5}$-AOD correlation, and PBLH may impose an adverse impact on the daily relationship in Yunnan province and weaken the seasonal relationship in summer in central China such as Jiangxi, Shaanxi, and Henan provinces. Compared with PBLH,

RH exerted a greater contribution to improving $PM_{2.5}$-AOD correlation in spring and had a weaker influence in winter. In addition, the correlation peaked at 400-600 m primarily in basin terrain such as Sichuan Basin, Shanxi-Shaanxi Basins and Junggar Basin. Nevertheless, despite of the publicly available high-resolution AOD data during the satellite era, an accurate reconstruction of high-resolution from satellite remote sensing remains greatly challenging, not only due to the uncertainties existing in the satellite aerosol retrievals but also due to the uncertainties coherent in $PM_{2.5}$ and AOD which co-vary with

topography and meteorological conditions. Even though the individual effect of humidity, vertical structure and terrain has been clarified, the synthetic impact of these factors has been recognized to be core to regulating the statistical relationship between near surface $PM_{2.5}$ and total column AOD. Therefore, more work in this regard is in urgent need. In addition, aerosol properties such as aerosol types are plausible factor that could modulate the $PM_{2.5}$-AOD relationship, which is worthy of more attention in the future.


**Data availability.** $PM_{2.5}$ measurements are from the official website of the Ministry of Ecology and Environment of China (https://www.mee.gov.cn). MAIAC aerosol data are available from NASA LAADS website at https://ladsweb.modaps.eosdis.nasa.gov/search/. AERONET data are publicly at https://aeronet.gsfc.nasa.gov/.

**Author Contributions.** Q. He and S.H.L Yim designed the whole experiment. Q. He developed the experiment code,

implemented and validated it. The manuscript was initially written by Q. He and revised by S.H.L Yim and M. Wang.

**Competing interests.** The authors declare that they have no conflict of interest.

**Acknowledgements.** We express our sincere appreciation to NASA for providing MAIAC and AERONET data.

**Finial support.** This research was funded by the Vice-Chancellor's Discretionary Fund of The Chinese University of Hong Kong (grant No. 4930744), Dr. Stanley Ho Medicine Development Foundation (grant No. 8305509) and the National

Natural Science Foundation of China (grant No. 41901324).

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





## Tables

**Table 1. Correlation coefficients between (RH-corrected) PM$_{2.5}$ and (PBLH-corrected) AOD over various urban agglomerations in China in the study period, including the Beijing-Tianjin-Hebei (BTH), Yangtze River Delta (YRD), Pearl River Delta (PRD), Triangle of Central China (TCC), Chengdu-Chongqing (CDCQ), central-and-southern Liaoning (CSL), Shandong Peninsula (SDP), Harbin-Changchun (HC), Central Plain (CP), Guanzhong Plain (GZP), and Northern Tianshan (NTS).**


| | Region | 2019 | Spring | Summer | Autumn | Winter | | Region | 2019 | Spring | Summer | Autumn | Winter |
|---|---|---|---|---|---|---|---|---|---|---|---|---|---|
| **Original data** | BTH | 0.53 | 0.75 | 0.53 | 0.70 | 0.62 | **Both corrected** | BTH | 0.70 | 0.80 | 0.60 | 0.77 | 0.66 |
| | YRD | 0.33 | 0.36 | 0.44 | 0.35 | 0.43 | | YRD | 0.62 | 0.51 | 0.54 | 0.67 | 0.55 |
| | PRD | 0.30 | 0.27 | 0.39 | 0.45 | 0.40 | | PRD | 0.62 | 0.25 | 0.44 | 0.66 | 0.73 |
| | TCC | 0.40 | 0.31 | 0.48 | 0.40 | 0.58 | | TCC | 0.72 | 0.52 | 0.52 | 0.65 | 0.64 |
| | CDCQ | 0.53 | 0.40 | 0.41 | 0.68 | 0.46 | | CDCQ | 0.64 | 0.41 | 0.41 | 0.55 | 0.62 |
| | CSL | 0.47 | 0.73 | 0.40 | 0.55 | 0.49 | | CSL | 0.69 | 0.81 | 0.47 | 0.66 | 0.73 |
| | SDP | 0.48 | 0.67 | 0.42 | 0.56 | 0.62 | | SDP | 0.71 | 0.74 | 0.56 | 0.72 | 0.71 |
| | HC | 0.31 | 0.52 | 0.34 | 0.53 | 0.47 | | HC | 0.59 | 0.68 | 0.47 | 0.71 | 0.61 |
| | GZP | 0.58 | 0.40 | 0.55 | 0.53 | 0.72 | | GZP | 0.83 | 0.58 | 0.69 | 0.76 | 0.83 |
| | CP | 0.47 | 0.67 | 0.54 | 0.55 | 0.61 | | CP | 0.76 | 0.79 | 0.63 | 0.79 | 0.70 |
| | NTS | 0.26 | 0.37 | 0.21 | 0.23 | -- | | NTS | 0.60 | 0.51 | 0.36 | 0.52 | -- |
| **RH-corrected PM$_{2.5}$** | BTH | 0.54 | 0.81 | 0.58 | 0.79 | 0.51 | **PBLH-corrected AOD** | BTH | 0.68 | 0.75 | 0.43 | 0.71 | 0.69 |
| | YRD | 0.43 | 0.54 | 0.46 | 0.51 | 0.52 | | YRD | 0.54 | 0.31 | 0.46 | 0.56 | 0.49 |
| | PRD | 0.50 | 0.34 | 0.44 | 0.55 | 0.57 | | PRD | 0.36 | 0.16 | 0.34 | 0.50 | 0.52 |
| | TCC | 0.45 | 0.38 | 0.49 | 0.47 | 0.59 | | TCC | 0.69 | 0.40 | 0.35 | 0.58 | 0.65 |
| | CDCQ | 0.50 | 0.41 | 0.38 | 0.52 | 0.58 | | CDCQ | 0.57 | 0.38 | 0.32 | 0.68 | 0.42 |
| | CSL | 0.58 | 0.76 | 0.48 | 0.69 | 0.55 | | CSL | 0.58 | 0.73 | 0.28 | 0.58 | 0.65 |
| | SDP | 0.54 | 0.73 | 0.47 | 0.64 | 0.63 | | SDP | 0.65 | 0.64 | 0.31 | 0.65 | 0.71 |
| | HC | 0.44 | 0.58 | 0.45 | 0.63 | 0.48 | | HC | 0.42 | 0.60 | 0.19 | 0.65 | 0.51 |
| | GZP | 0.62 | 0.51 | 0.60 | 0.58 | 0.72 | | GZP | 0.78 | 0.42 | 0.43 | 0.71 | 0.79 |
| | CP | 0.53 | 0.68 | 0.60 | 0.66 | 0.59 | | CP | 0.71 | 0.73 | 0.42 | 0.71 | 0.68 |
| | NTS | 0.30 | 0.44 | 0.28 | 0.26 | -- | | NTS | 0.49 | 0.42 | 0.29 | 0.45 | -- |





**Figures**

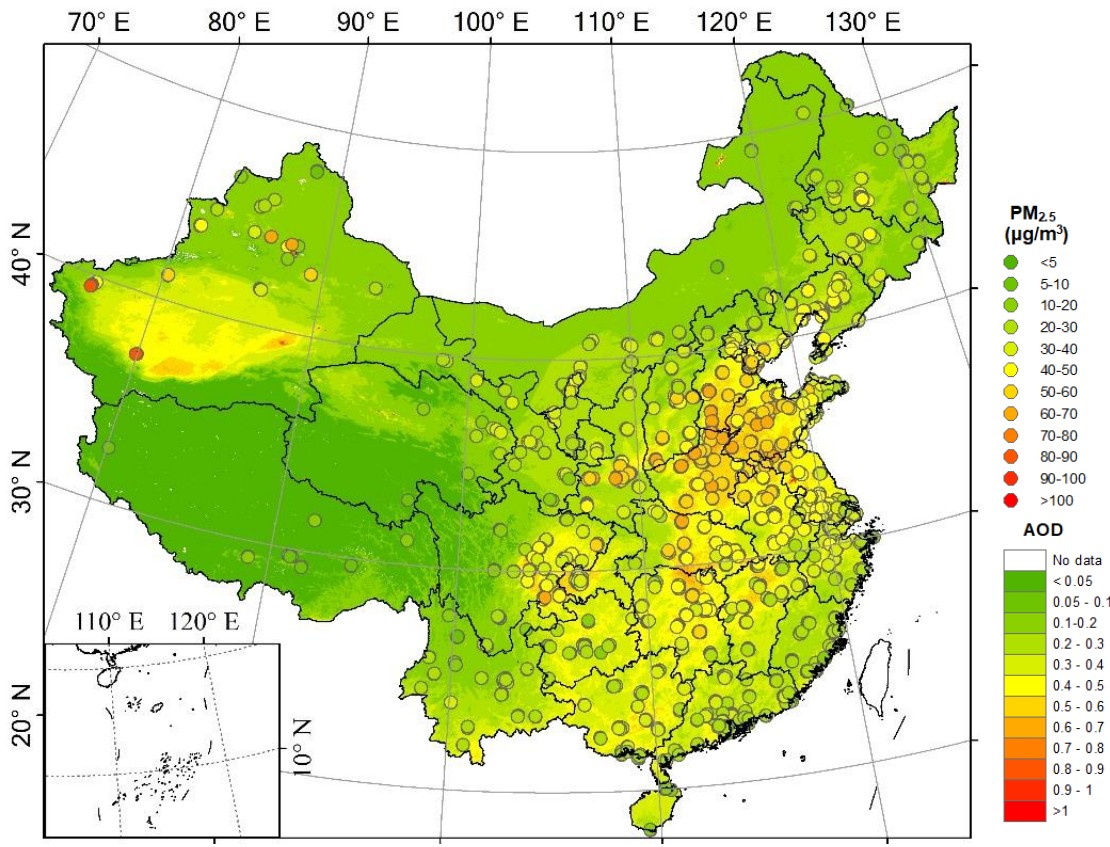

**Figure 1. The spatial distribution of ground-level PM₂.₅ and satellite AOD averages in China in 2019. The mean value of ground-level PM₂.₅ concentration was calculated from all measured days which covers almost the whole study period while the mean AOD value comes from less days because the MAIAC algorithm could not make retrievals over areas with high cloud coverage.**

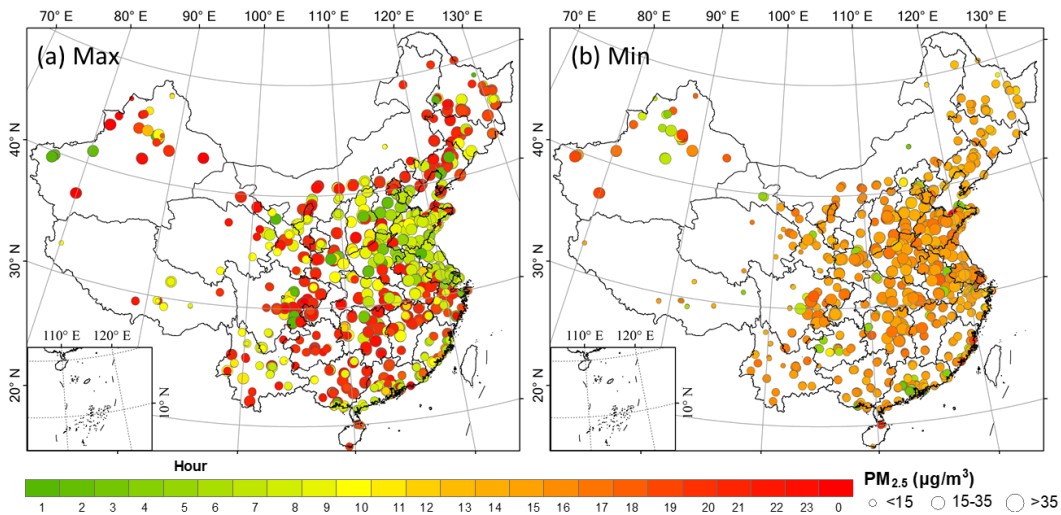

**Figure 2. Spatial distribution of diurnal PM$_{2.5}$ averaged over the 2019 period according to (a) maximum and (b) minimum mean PM$_{2.5}$ concentration in the 24 h. The solid color and size of the circle denote the hour with maximum and minimum diurnal mean concentration and the corresponding concentration value over each monitoring station, respectively.**


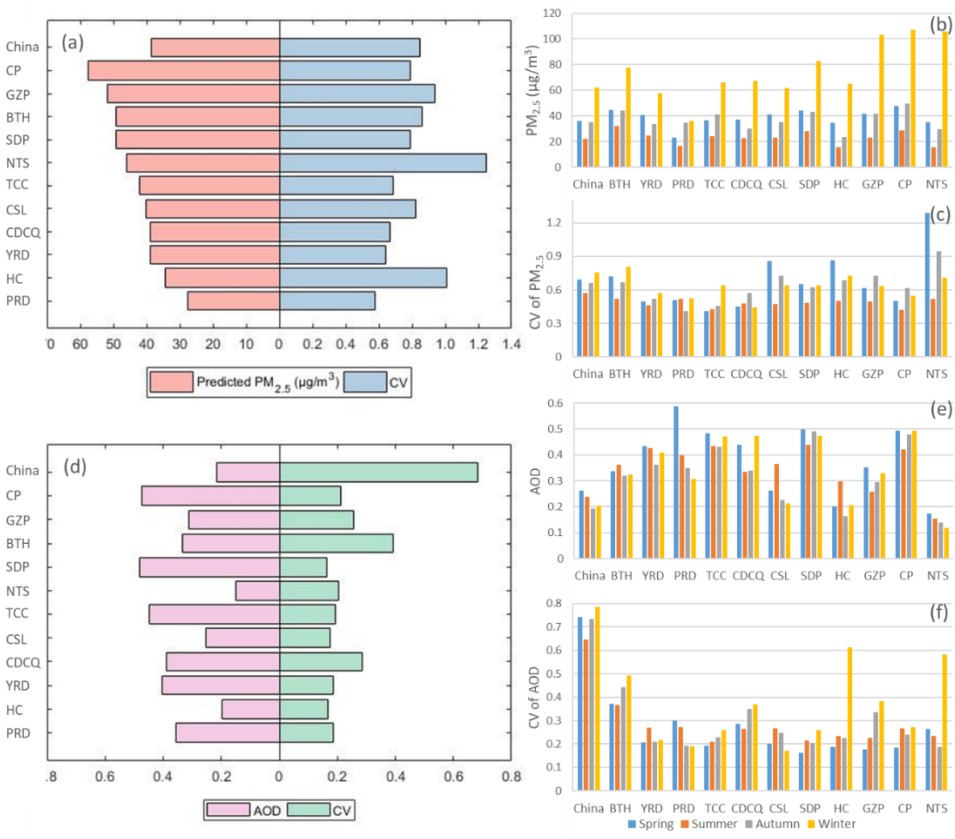

**Figure 3. Regional mean values and coefficients of variation (CV) of PM₂.₅ and AOD in 2019. (a) is the mean PM₂.₅ and its CV for eleven urban agglomerations in the whole study period, and (b) and (c) is the mean PM₂.₅ and the CV value in each season, respectively; (c) is the mean AOD and its CV in regions during the whole year, and (e) and (f) is the mean AOD and the CV value in each season, respectively.**


**Figure 4. Spatial distributions of relationship between ambient PM₂.₅ and satellite AOD collocations over China in 2019. (a-e) refers to the correlation coefficient during all years and each season, respectively. (f-j) are based on the slope values of linear regression model. Please note that only statistically significant results, i.e., p-level<0.05, are presented here.**





**Figure 5. Scatter plots for the paired PM$_{2.5}$ concentration (μg/m3) and satellite AOD over the eleven urban agglomerations in China including Beijing-Tianjin-Hebei (BTH), Yangtze River Delta (YRD), Pearl River Delta (PRD), Triangle of Central China (TCC), Chengdu-Chongqing (CDCQ), central-and-southern Liaoning (CSL), Shandong Peninsula (SDP), Harbin-Changchun (HC), Central Plain (CP), Guanzhong Plain (GZP), and Northern Tianshan (NTS). The solid line represents the linear trend.**




**Figure 6. The relationship between PM₂.₅ and AOD in China in the entire year of 2019 and in each season based on the original PM₂.₅ and AOD observations (left column), only PM₂.₅ corrected by RH (middle-left column), only AOD corrected by PBLH (middle right column) and both corrected data (right column). Please note that only stations with statistical significance (i.e., p-level<0.5) are presented in this figure.**




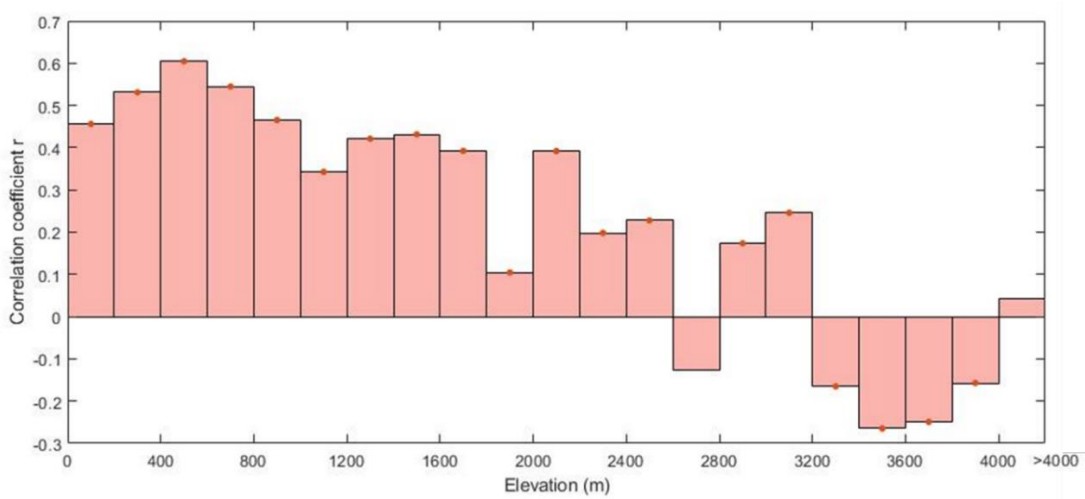

**Figure 7. Correlation coefficients of PM$_{2.5}$-AOD pairs plotted against binned elevations. Note that the red asterisk indicates**
**statistically significant level at p<0.05.**