# Peer review of "The spatiotemporal relationship between PM2.5 and AOD in China: Influencing factors and Implications for satellite PM2.5 estimations by MAIAC AOD"

_Atmospheric Chemistry and Physics, 2021_

## Author Comment (AC1)

Dear Reviewer,

Thank you very much for your effort and comments. We addressed them one by one below. Hope you find our revisions useful. Thank you again.

Yours faithfully,

Steve
Associate Professor
Department of Geography and Resource Management

The Chinese University of Hong Kong, Shatin, Hong Kong
Tel: (852) 3943 6534
Fax: (852) 2603 5006
Email: steveyim@cuhk.edu.hk
GRMD@CUHK: http://www.grm.cuhk.edu.hk/eng/

**This manuscript studied the relationship between surface PM2.5 and AOD, and also the influence of meteorology and topography on the relationship in high spatial resolution. The topic is critical the improved knowledge can contribute to the research field, enhancing air pollution research using satellite-retrieval technology. I suggest to accept this work for publication after revisions.**

**(1)      Title: AOD should be replaced by aerosol optical depth.**
**Response:** Thanks for your suggestion. We have replaced AOD with aerosol optical depth in the revised manuscript.

**(2)      Line 159: the word "vs." should be revised.**
**Response:** We have replaced "vs." by "and".

**(3)      Table 1: Why the result is missing in Northern Tianshan in winter? Is it because cloudy weather over there during the season?**
**Response:** In Northern Tianshan in winter, PM2.5-AOD pairs are not available. This is partly due to cloud contamination and the limitation of retrieval algorithm such that a few of AOD retrievals have been provided during that season. In addition, it is also partly because there is a limited number of monitoring sites over this region.

**(4)      Figure 1: the texts in the legends are small. They should be enlarged.**
**Response**: We have enlarged the texts in the legends in the revised version.

**(5)      Line 623: The figure should be the spatial distribution of annual mean ground-level PM2.5 and satellite AOD in China in 2019. It is better to add "annual mean" to make the meaning clearer.**
**Response:** Correct, thanks for your suggestion. We have added "annual mean" in the caption of Figure 1.

**(6)      Figure 2: the texts of the legend are too small.**

**Response**: We have enlarged the texts of the legend in the revised version.

**(7)     Line 174: What is the reason about the U-shape trend? Why is the PM2.5 concentration lower during Jun-Aug?**
**Response:** The PM2.5 concentration is generally higher in winter and lower in summer. The higher value in wintertime is related to relatively higher human air pollutant emissions (e.g., more coal and biomass burning for heating in winter) and unfavorable meteorological conditions for pollution dispersion (e.g., lower boundary layer in stable atmosphere). In contrast, the lower value in summertime is probably related to lower human air pollutant emissions (e.g., reduced coal and biomass burning for heating during summer) and favorable meteorological conditions for pollution dispersion (e.g., enhanced convection and more precipitation during summer). We have added the discussion in the second paragraph of Section 3.1.

**(8)     Line 188: the sum of the stations (559+519+50) is not equal to 1494. Please check the number of the stations.**
**Response**: Thanks for your question. The values are correct; nevertheless, the writing is confusing. The content is revised to avoid confusion.

**Revised version:**
Among the 1494 monitoring stations, the largest diurnal PM2.5 concentration appeared in the evening (19:00-23:00) at 638 sites (~42.70%), followed by 567 sites (~37.95%) with the maximum concentration in morning (7:00-11:00). There were only 50 sites (~3.35%) reaching its maximum in afternoon (12:00-18:00), whereas the remaining 239 sites (~16.00%) had their maximum in midnight (00:00-06:00).

**(9)     Line 171: It should be explicitly mentioned Figure S3 is shown in supplementary information.**
**Response**: We have clarified that Figure S3 is located in supplementary information in the revised manuscript.

**(10)     Figure S9 in supplementary information: the word of "elevatior" should be "elevation" The caption of Figure S9: the second sentence should be revised: Note that the asterisk after r values represent statistically insignificance (p>0.05).**
**Response**: We have revised the word "elevation" and the caption for the figure.

---

## Author Comment (AC2)

Dear Reviewer,

Thank you very much for your effort and comments. We addressed them one by one below. Hope you find our revisions useful. Thank you again.

Yours faithfully,

Steve
Associate Professor
Department of Geography and Resource Management

The Chinese University of Hong Kong, Shatin, Hong Kong
Tel: (852) 3943 6534
Fax: (852) 2603 5006
Email: steveyim@cuhk.edu.hk
GRMD@CUHK: http://www.grm.cuhk.edu.hk/eng/

**Satellite-based retrieval of PM2.5 shows great advantage over traditional ground-level observations, which largely relies on the well-established statistical relationship between AOD and PM2.5. This manuscript by He et al. investigated the relationship between PM2.5 and AOD in China using the high-resolution AOD products from MAIAC, and the major influencing factors, including relative humidity, PBLH, and terrain, have been discussed. Overall, this work is well organized, and the analysis methods are scientifically sound. I am almost in a position to accept it for publication at ACP after the authors adequately address the following concerns, even though most of them are minor.**

**Specific comments:**
**(1)      L11: "to monitoring surface PM2.5 concentration" can be revised to "surface-based PM2.5 concentration observations"**
**Response**: Thanks for your suggestion. We have revised the description in the revised manuscript.

**(2)      L13: "ground" -> "ground-level "**
**Response**: We have replaced "ground" with "ground-level" in the revised version.

**(3)      L29: the article is missing before "Findings"**
**Response**: The article has been added at the beginning of the sentence.

**(4)      L38-43: This is a pretty long sentence and can be shortened or rephrased.**
**Response**: This long sentence has been replaced with two short sentences.

**(5)      L47: "heavily" -> "heavy"**
**Response**: We has revised this word.

**(6)      L52: "tuning" needs to be corrected.**

**Response**: "tuning" has been revised as "concordant".

**(7)   L62-63: "further exploring the relationship between near-surface PM2.5 and MAIAC AOD for China during a long-term period is necessary." can be rephrased as "further exploring the relationship seems critically imperative between near-surface PM2.5 and MAIAC AOD over China based on much finer AOD products."**
**Response**: We have revised this sentence accordingly.

**(8)   L65: "suggested" -> "recognized"; also, "different definitions" can be revised to "the definition difference"**
**Response**: We have revised them based on your suggestion.

**(9)   L70: "The vertical structure" of aerosols? Please clarify it.**
**Response**: It is the vertical structure of aerosols. We have clarified it in the revised manuscript.

**(10)   L80: The reference of Guo et al. 2017 is wrongly cited here, since the effect of aerosol diurnal variability on precipitation did not involve at all. Instead, the author may refer to Kim et al. 2010 (doi: 10.1007/s00382-010-0750-1); Guo et al., 2016 (doi:10.1002/2015JD023257) and Lee et al. 2016 (doi: 10.1002/2015JD024362), and Zheng et al. 2020 (doi: 10.1088/1748-9326/ab99fc), among others.**
**Response**: Thanks for your suggestion. We have revised the references.

**(11)   L303-309: Except for the different number of available observations for AOD and PM2.5, the main causes for the mismatch between AOD and PM2.5 lie at the fundamental discrepancy in physical concept. AOD is a unitless variable that denotes the total extinction induced by aerosol in the whole atmospheric column, whereas PM2.5 represents the aerosol concentration measured at the ground level.**
**Response**: Thanks for your suggestion. We have added the definition difference between PM2.5 and AOD as the main cause for the PM2.5-AOD mismatch.

**(12)   L378-379: The diurnal variability of PBLH needs to be specified from a climatological perspective, and the authors are suggested to refer to Liu and Liang 2010 (doi:10.1175/2010JCLI3552.1); Zhang et al. 2018 (doi: 10.1175/JCLI-D-17-0231.1)**
**Response**: We have revised this sentence by following your suggestion.

**(13)   L395: "by PBLH correction" -> "corrected for PBLH"**
**Response**: We have revised this sentence as per your suggestion.

**(14)   L399: "of " is redundant.**
**Response**: "Of" in this sentence has been removed.

**(15)   L421: "Corresponding to" -> "Generally consistent with"**
**Response**: We have revised this sentence by following your suggestion.

**(16)    L455: "As far as the meteorology" -> "As far as the confounding impact of meteorology"**
**Response**: We have revised this sentence as per your suggestion.

**(17)    L460: "despite of" -> "despite"**
**Response**: "Despite of" has been replaced by "despite".

**(18)    L465: "The vertical structure" of aerosols? Please clarify it.**
**Response**: It is the vertical structure of aerosols. We have clarified it in the revised manuscript.

**(19)    L465-466: "Even though the individual effect…., the synthetic impact of these factors has been recognized to be core…" seems problematic in terms of logic. The joint effect of these factors could be revealed using the multi-regression analysis in the future. This point is suggested to be taken into account when revising this sentence.**
**Response**: We have revised this sentence accordingly.

**(20)    The variables appeared in Eqs. 1-2 should be clarified, including the units.**
**Response**: We have explained the variables in Eqs.1-2 in the revised version.

**(21)    The quality of Figure 6 is a little low for its blurred x-axis and y-axis titles, and I strongly recommend the authors redraw it.**
**Response**: We have replotted this figure to enhance its quality.

---

## Author Comment (AC3)

Dear Reviewer,

Thank you very much for your effort and comments. We addressed them one by one below. Hope you find our revisions useful. Thank you again.

Yours faithfully,

Steve
Associate Professor
Department of Geography and Resource Management

The Chinese University of Hong Kong, Shatin, Hong Kong
Tel: (852) 3943 6534
Fax: (852) 2603 5006
Email: steveyim@cuhk.edu.hk
GRMD@CUHK: http://www.grm.cuhk.edu.hk/eng/

**The authors investigated the spatiotemporal relationship between ground-level PM2.5 measurements and satellite derived AOD data over China for the year of 2019. Compared with previous research with similar topic, this study used high-spatial-resolution AOD data with combination of AERONET data which provide high-temporal-resolution aerosol data, which is important for understanding the relationship of PM2.5-AOD and useful for ground-level PM2.5 estimation, especially when deriving PM2.5 from satellite remote sensing data is becomes more popular. The findings of this study are worth of publication in the journal after minor revision as followings:**

**(1)     My major concern is the way that you matched PM2.5 and AOD in space, that isï¼Œ what is the radius of buffer zone around the site for AOD average? please clarify it in detail.**
**Response**: Thanks for your question. we used the 1-km grid of AOD to match the in-situ $PM_{2.5}$ measurements and averaged those $PM_{2.5}$ observations from multiple monitors located within a 1-km grid cell. We have clarified the data integration method at the first paragraph of Section 2.3.

**(2)     Line 39-44: A long sentence. Please rewrite it.**
**Response**: This long sentence has been replaced with two short sentences in the revised manuscript.

**(3)     Line 108: It seems no Fig. A1 in supporting document. May be Fig. S1?**
**Response**: It has been updated by "Figure S1" in the revised version.

**(4)     Line 125-128: Please clarify which level of AERONET data were used in this study.**
**Response**: AERONET Level 2.0 data were used in this study. We have clarified it in the revised version.

**(5)   1 and 2: Please explicitly describe the two equations. What is the meaning for each variable?**

**Response**: We have provided the explanation for each variable in the two equations.

**(6)   Line 344: "pollutant acumination"? May be "accumulation".**

**Response**: Yes, it is correct. We have revised this word in the revised manuscript.

**(7)   Table 1: Why there is no value for NTS in winter?**

**Response:** In Northern Tianshan in winter, PM2.5-AOD pairs are not available. This is partly due to cloud contamination and the limitation of retrieval algorithm such that a few of AOD retrievals have been provided during that season. In addition, it is also partly because there is a limited number of monitoring sites over this region.

**(8)   Figure 2: the text of "hour" legend is a little small.**

**Response**: We have enlarged the texts of legends for this figure.

**(9)   In figure 5, suggest that the count is replaced by frequency, which can be easily compared among different regions, due to their different samples.**

**Response**: We have replotted this figure as per your suggestion.

**(10)   Figure 6: The text for latitude and longitude is a little small. Please replot it.**

**Response**: We have redrawn Figure 6 in the revised version.

**(11)   Figure S9: it should be "elevation"?**

**Response**: Yes, it is correct. We have revised this word in the revised manuscript.